# Epidemiological survey of serum titers from adults against various Gram-negative bacterial V-antigens

Mao Kinoshita[1], Masaru Shimizu[1], Koichi Akiyama[1], Hideya Kato[1], Kiyoshi Moriyama[2], Teiji Sawa[1] *

1 Department of Anesthesiology, Kyoto Prefectural University of Medicine, Kyoto, Japan, 2 Department of Anesthesiology, School of Medicine, Kyorin University, Tokyo, Japan

* anesth@koto.kpu-m.ac.jp

**Data Availability Statement:** All relevant data are within the paper.

**Funding:** This work was supported by grants from Grant-in-Aid for Scientific Research (KAKENHI No.

## Abstract

The V-antigen, a virulence-associated protein, was first identified in *Yersinia pestis* more than half a century ago. Since then, other V-antigen homologs and orthologs have been discovered and are now considered as critical molecules for the toxic effects mediated by the type III secretion system during infections caused by various pathogenic Gram-negative bacteria. After purifying recombinant V-antigen proteins, including PcrV from *Pseudomonas aeruginosa*, LcrV from *Yersinia*, LssV from *Photorhabdus luminescens*, AcrV from *Aeromonas salmonicida*, and VcrV from *Vibrio parahaemolyticus*, we developed an enzyme-linked immunoabsorbent assay to measure titers against each V-antigen in sera collected from 186 adult volunteers. Different titer-specific correlation levels were determined for the five V-antigens. The anti-LcrV and anti-AcrV titers shared the highest correlation with each other with a correlation coefficient of 0.84. The next highest correlation coefficient was between anti-AcrV and anti-VcrV titers at 0.79, while the lowest correlation was found between anti-LcrV and anti-VcrV titers, which were still higher than 0.7. Sera from mice immunized with one of the five recombinant V-antigens displayed cross-antigenicity with some of the other four V-antigens, supporting the results from the human sera. Thus, the serum anti-V-antigen titer measurement system may be used for epidemiological investigations of various pathogenic Gram-negative bacteria.

## Introduction

The type III secretion system (TTSS) plays a major role in the virulence of many Gram-negative bacteria [1–3]. Through the TTSS, Gram-negative bacteria inject their effector molecules to target eukaryotic cells and induce a favorable environment for their infections. Translocation is a mechanism through which effector molecules of the TTSS pass through the targeted eukaryotic cell membrane. During translocation, three bacterial proteins form a translocational pore structure called the 'translocon' [1–3]. A cap protein in the secretion apparatus of the type III secretion needle is one type of translocon protein, which is called the V-antigen in *Yersinia* spp. for historical reasons [4–8]. Briefly, the V-antigen, a virulence-associated protein,

24390403, 26670791, 18H02905, 20816384, and 18K16521) and the Ministry of Education, Culture, Sports, Science and Technology of Japan to Teiji Sawa.

**Competing interests:** T. Sawa has patents associated with PcrV immunization (World Patent No. WO0033872; European Patent No. EP1049488; U.S. Patent No. 6309651; U.S. Patent No. 6827935, and Japan Patent No. 2017-020501). Until 2011, T. Sawa received a patent fee from the Regents of the University of California related to the development of a therapeutic monoclonal antibody at KaloBios Pharmaceutical. This does not alter our adherence to PLOS ONE policies on sharing data and materials. There are no current financial relationships with any organization associated with this study.

was identified as an antigenic component recognized by the immune system in *Yersinia pestis* plague-infected hosts more than half a century ago [4–8]. In the 1980s, the V-antigen of *Y. pestis* was identified as a low calcium response (*lcr*)-associated protein (named LcrV) encoded in the plasmid associated with its virulence [9]. A homologous gene called PcrV was identified in the *Pseudomonas aeruginosa* genome [10]. It has been reported that the virulence-associated with the TTSS can be inhibited by a specific antibody against LcrV in *Yersinia* and PcrV in *P. aeruginosa* [11, 12]. Because vaccinating mice with LcrV or PcrV has protective effects against lethal infections by *Yersinia* or *P. aeruginosa*, respectively, anti-PcrV immunotherapies were developed to target human infections with *P. aeruginosa* using immunoglobulins [13–24] and vaccines [25–27], from which several projects have progressed to human clinical trials [28–31].

We recently published an epidemiological study on serum titers against PcrV in human volunteers [32], and another showing how prophylactic administration of human serum-derived immunoglobulin with a high anti-PcrV titer significantly improves the survival rate, pulmonary edema, and inflammatory cytokine production of a *P. aeruginosa* pneumonia model [18]. The results of both studies imply that immunity against the V-antigen and its homologs might be necessary to prevent infections caused by pathogenic bacterial species employing the TTSS-virulence mechanism [18, 32]. V-antigen homologs have been recently reported in several Gram-negative bacteria, including *Aeromonas* spp., *Vibrio* spp., and *Photorhabdus luminescens* (hereafter referred to as *Ph. luminescens*) [33]. Although specific immunity against the V-antigen or its homologs appears to be important for host immunity against such bacterial infections, insufficient information is available on human immunity against V-antigens. Therefore, here, we conducted an epidemiological study on serum titers against the V antigen and its homologs in *Y. pestis*, *Aeromonas salmonicida*, *Vibrio parahaemolyticus*, and *Ph. luminescens*. Potential associations in terms of age, titer levels, and cross-reactivity were evaluated among the recombinant V-antigen homologs. Moreover, for some species, the titer levels against these antigens were highly correlated, and some V-antigen homologs showed cross-reactivity.

## Materials and methods

### Construction of five recombinant Gram-negative bacterial V-antigens (PcrV, LcrV, AcrV, VcrV, and LssV) and *P. aeruginosa* porin F from the outer membrane (OprF)

Five recombinant V-antigens and recombinant *P. aeruginosa* OprF were constructed. Details on the PCR primers and cloning sites are listed in **Table 1**. The coding regions of the V-antigens were amplified by polymerase chain reaction (PCR) with specific primers containing restriction enzyme sites for insertion into a protein expression vector. PCR-amplified genes were cloned into the pCR2.1 cloning vector and *E. coli* TOP10F cells via TOPO cloning (Thermo Fisher Scientific, Waltham, MA, USA). After digesting the purified plasmids containing each individual cloned gene with restriction enzymes, the inserted coding regions of each gene were transferred to the multiple cloning site of the expression vector pQE30 (Qiagen, Hilden, Germany) for expression of a hexahistidine-tagged protein in *E. coli* M15. The various endotoxin-free Gram-negative bacteria V-antigens were prepared as reported previously (**Fig 1**) [17, 20].

### Survey participants and study background

This study was approved by the Ethics Committee of Kyoto Prefectural University of Medicine (Approval number: RBMR-E-326-1; Kyoto, Japan). As a non-interventional and non-invasive retrospective observational study, the need for consent was waived by the ethics committee.

**Table 1. Gene sources, primer sets for V-antigen and OprF gene cloning, and characteristics of the recombinant V-antigens and OprF used in this study.**

| Gene | Gene source | Restriction enzyme site for expression vector | Cloning PCR primers* | Coding region size (bp) | Protein | Amino-acids | MW (kDa) | Ref |
|---|---|---|---|---|---|---|---|---|
| pcrV | Pseudomonas aeruginosa PA103 | BamHI HindIII | 5'-CGGGATCCATGGAAGTCAGAAACTTTAA-3' 5'-AAGCTTCTAGATCGCGCTGAGAATGT-3' | 927 | PcrV | 306 | 33.8 | [34–36] |
| lcrV | Yersinia pestis pCD1 plasmid | BamHI HindIII | 5'-CGGGATCCATGATTAGAGCCTACGAACAAAACCCACAA-3' 5'-AAGCTTTCATTTACCAGACGTGTCATCTAGCAGACG-3' | 1,023 | LcrV | 338 | 38.6 | [9] |
| acrV | Plasmid JF2267 of Aeromonas salmonicida subsp. Salmonicida, | SphI SalI | 5'-GCATGCATGAGCACAATCCCTGACTAC-3' 5'-GTCGACTCAAATTGCGCCAAGAATGTCG-3' | 1,134 | AcrV | 375 | 41.6 | 14 |
| vcrV | Vibrio parahaemolyticus, ATCC 17802D-5 | BamHI PstI | 5'-CGGGATCCATGACGGATATGACAACAAC-3' 5'-CTGCAGTTAAATGGCTCGTAGGATTTCTTG-3' | 1,866 | VcrV | 619 | 68.3 | - |
| lssV | Photorhabdus luminescens subsp, luminescens, ATCC 29999 | BamHI HindIII | 5'-GGATCCATGGAAATAGGCCATATCAAA-3' 5'-AAGCTTTTAAATTGCGCCGAGAATAT-3' | 1,020 | LssV | 337 | 38.3 | - |
| oprF | Pseudomonas aeruginosa PAO1 | BamHI HindIII | 5'-CGGGATCCATGAAACTGAAGAACACCTTAG -3' 5'-AAGCTTTTACTTGGCTTC AGCTTCTAC-3' | 1,053 | OprF | 362 | 39.0 | - |

*Underlines indicate restriction enzyme sites

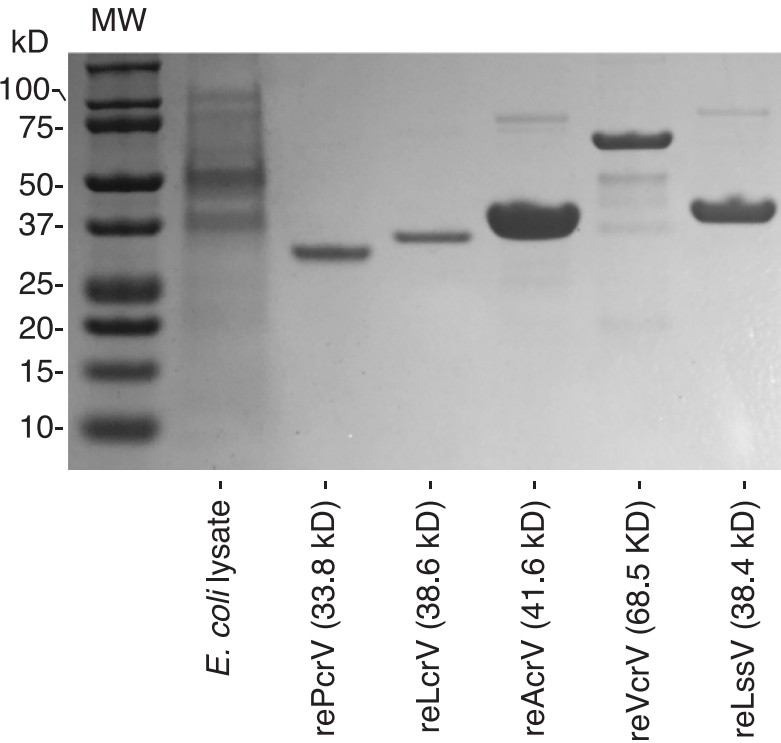

**Fig 1. Sodium dodecyl sulfate-polyacrylamide gel electrophoresis (SDS-PAGE) separation of extracted recombinant hexahistidine-tagged V-antigen proteins.** Recombinant PcrV from *P. aeruginosa*, LcrV from *Y. pestis*, AcrV from *Aeromonas salmonicida*, VcrV from *V. parahaemolyticus*, and LssV from *Ph. luminescens* were separated by SDS-PAGE using a 10% Bis-Tris-gel.

Adult patients (n = 186) who underwent anesthesia in the Central Operating Division of Kyoto Prefectural University of Medicine, from April 2012 to March 2013, participated in this study as volunteers, as reported previously [32]. Briefly, serum was prepared from the remaining small amount of each blood sample collected for arterial blood gas analysis for induction of anesthesia and then stored at −80˚C.

## Anti-V antigen titer measurements

We developed an enzyme-linked immunoabsorbent assay (ELISA) to measure serum anti-V antigen titers. Microwell plates (Nunc C96 Maxisorp; Thermo Fisher Scientific) were coated for 2 h at 4˚C with recombinant V-antigen proteins (recombinant PcrV from *P. aeruginosa*, LcrV from *Yersinia*, AcrV from *Aeromonas*, VcrV from *Vibrio*, and LssV from *Ph. luminescens*) suspended in coating buffer (1 μg/mL in coating solution; 0.05 M NaHCO$_3$, pH 9.6) [32] (**Fig 1**). The plates were washed twice with phosphate-buffered saline (PBS) containing 0.05% Tween-20 (P9416; Sigma-Aldrich, St. Louis, MO, USA) and then blocked with 200 μL of 1% bovine serum albumin/PBS overnight at 4˚C. Samples (serial dilution: 1280×) were applied to the plates (100 μL/well) and then incubated overnight at 4˚C. Peroxidase-labeled anti-human IgG (A8667, Sigma- Aldrich) was applied at 1:60,000 for 1 h at 37˚C. After six washes, the plates were incubated with 2,2'-azino-bis (3-ethylbenzthiazoline-6-sulfonic acid) (A3219; Sigma-Aldrich) at room temperature for 30 min. After adding 0.5 M H$_2$SO$_4$ at 100 μL/well to the plates, the optical density (OD) at 450 nm was measured with a microplate reader (MTP-880Lab; Corona Electric, Hitachinaka, Japan). To ensure no cross-reactivity with *Escherichia coli* proteins, we performed an inhibition ELISA with a soluble fraction of *E. coli* M15 lysate, and no significant effect on titer measurement was observed. Except for human monoclonal anti-PcrV IgG mAb 6F5 as a standard to measure the anti-PcrV titer [32], there is no human anti-V-antigen IgG. Therefore, after optimization of the ELISA system for anti-PcrV titers using the mAb 6F5 standard [32], the OD measured under a consistent condition with the same secondary antibody was used to evaluate the titers.

## Inhibition ELISA

Cross-reactivity was analyzed by an inhibition ELISA. Two human sera with relatively high anti-PcrV, LcrV, AcrV, VcrV, and LssV titers were diluted at 1:1000 and preincubated with either recombinant LssV or recombinant OprF (100 μg/mL) overnight at 4˚C. The anti-V-antigen titers were measured in triplicate by an ELISA using recombinant V-antigen-coated plates.

## Immunizing mice with the V-antigens

Certified pathogen-free, male ICR mice (4 weeks old) were purchased from Shimizu Laboratory Supplies, Co, Ltd (Kyoto, Japan). Mice were housed in cages with filter tops under pathogen-free conditions. The protocols for all animal experiments were approved by the Animal Research Committee of Kyoto Prefectural University of Medicine before undertaking the experiments (Authorization number: M29-592). Three mice per group were intradermally immunized with one of the five recombinant V-antigen proteins (10 μg/dose) adjuvanted with complete Freund's adjuvant in the first injection, and four weeks later with incomplete Freund's adjuvant for the second injection. Eight weeks after the first injection, the immunized mice were euthanized with a large dose intraperitoneal injection of sodium pentobarbital, and peripheral blood samples were collected. Serum titers against the five V-antigens were individually measured by ELISAs, as described above.

## Phylogenetic and cluster analyses

The five V-antigens were phylogenetically analyzed using ClustalW (Genome Net, https://www.genome.jp/tools-bin/clustalw) or RStudio (version 1.2, RStudio, Boston, MA, USA. https://www.rstudio.com) with R version 3.6.1 (The R Foundation, https://www.r-project.org). Unrooted trees were prepared using the neighbor-joining method, and rooted trees were prepared using the unweighted pair group method with arithmetic means applied to the ClustalW site and the standard R function *plot.hclust*. Heat maps where the individual values contained in a matrix are represented as colors with dendrograms prepared using the function *phylo of* the package "ape" for phylogenies (http://ape-package.ird.fr.) and the function *heatmap.2* of the package gplots. The predicted three-dimensional structures were generated with the Cn3D macromolecular structure viewer at the National Center for Biotechnology Information (https://www.ncbi.nlm.nih.gov/Structure/CN3D/cn3d.shtml).

## Statistical analysis

We performed a regression analysis of the measured antibody titers using the regression data analysis tool of Microsoft Excel for Mac (ver. 16.16.5, Microsoft Co., Redmond, WA, USA). Statistical analyses were conducted using InStat version 4.0 (GraphPad Software Inc., La Jolla, CA, USA). *P*-values were calculated using the Mann-Whitney U-test. A *p*-value of less than 0.05 was considered statistically significant.

# Results

## Anti-V antigen titers and volunteer age distribution

The 186 volunteers included 111 (59.7%) males and 75 (40.3%) females and an age distribution of 20–29 years (13, 7.0%), 30–39 years (20, 10.8%), 40–49 years (26, 14.0%), 50–59 years (23, 12.4%), 60–69 years (40, 21.5%), 70–79 years (42, 22.5%), and ≥80 years (22, 11.8%). No study participant had an active infection. The titers against the V-antigen across the age distribution of the 186 participants are shown in **Fig 2**. There was no statistically significant correlation in the linear regression between age and each anti-V antigen titer, although high anti-PcrV titers were more common at over 50 years of age in the population, as reported previously [32]. As an overall trend, two separate titer peaks in the 40–49 and 70–79 age groups were observed in the age distribution of all anti-V-antigen titers.

## Correlations among the anti-V antigen titers

We next analyzed whether any correlations existed among the five anti-V antigen titers. The anti-LcrV and the anti-AcrV titers showed the highest correlation with a correlation coefficient of 0.84, followed by the anti-AcrV and anti-VcrV titers at 0.79, and the anti-LcrV titers and anti-VcrV titers at 0.74 (**Fig 3**) Low correlations were detected between anti-PcrV and any of the other anti-V-antigens.

Next, the inhibition ELISA was performed to show cross-antigenicity among V-antigens. As an irrelevant non-crossreactive protein, we prepared *P. aeruginosa* recombinant OprF using the same *E.coli*-derived recombinant protein construction system. The sequence alignment scores obtained from ClustalW between OprF and V-antigens were 9.8–11.4, whereas those among V-antigens were from 21.3 (between VcrV and AcrV) to 48.3 (between LcrV and LssV). Two human sera diluted 1000× were preincubated with either recombinant LssV or recombinant OprF overnight. Then, preprocessed serum titers against each V-antigen were measured in comparison with the titer levels of the original sera (**Fig 5**). As a result, preincubation with OrpF did not affect the specific titer levels. However, preincubation with LssV

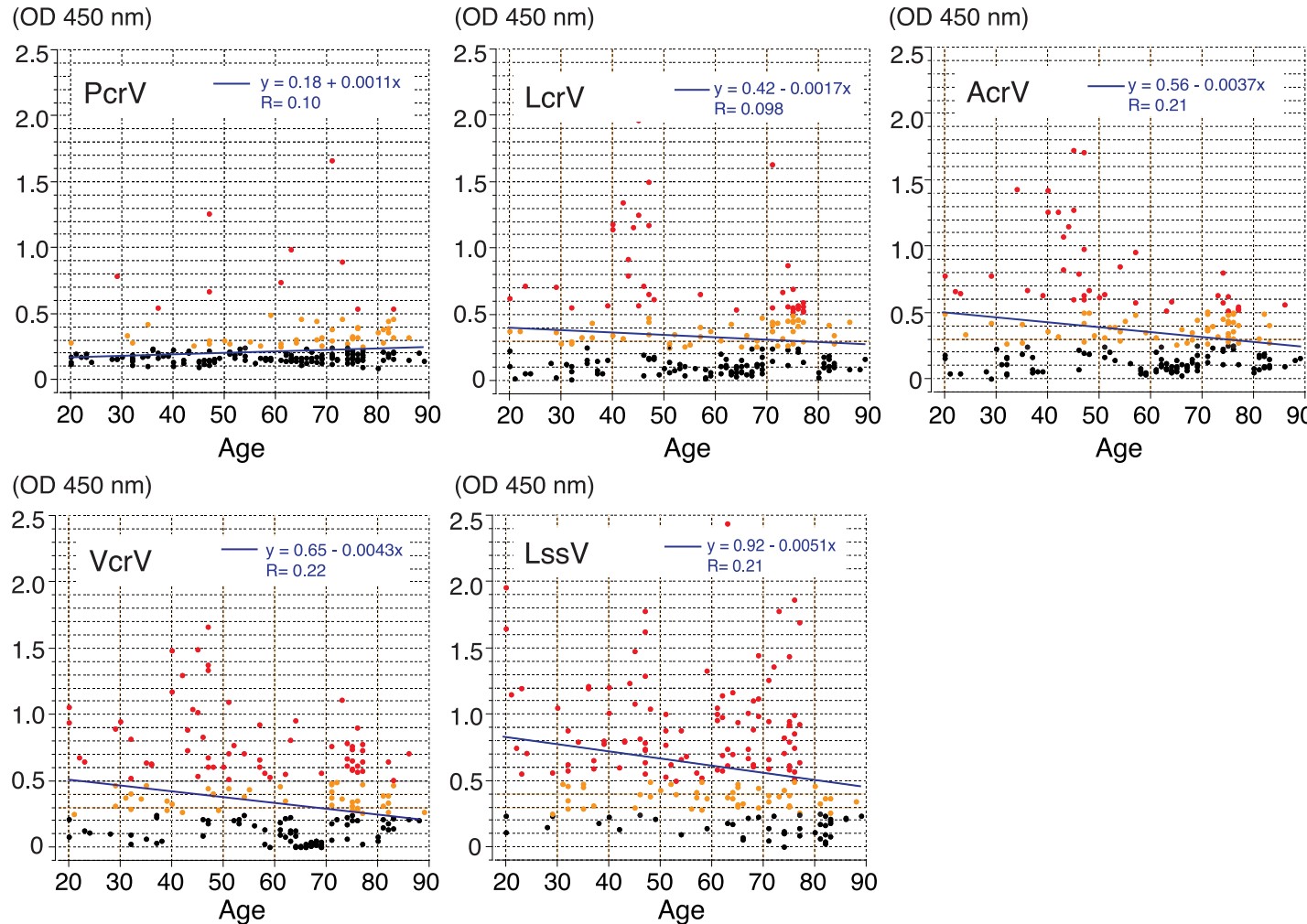

**Fig 2. Age distribution of human V-antigen titers.** The diluted serum (1,280×) was applied to ELISAs, and OD 450 nm values were measured. OD values of ≥0.5 are indicated by red dots, while yellow dots represent values between 0 and 0.3. OD: Optical density.

decreased the titer levels compared with the titer levels of the original sera (*$p < 0.05$ for AcrV, VcrV, and LssV).

Cluster analysis of the correlation coefficient values was conducted, and phylogenetic trees and a heat map were constructed (**Fig 4A**). The unrooted and rooted phylogenic trees and heat map showed that anti-PcrV had a unique profile among the five anti-V-antigen titers. The anti-LssV titer was located between anti-PcrV and the three other anti-V-antigen titers. Higher homology in terms of antigenicity was observed among anti-AcrV, anti-LcrV, and anti-VcrV titers.

## Cross-reactive antibodies against V-antigens and anti-V antigen titers in serum from immunized mice

To evaluate potential cross-antigenicity among the five V-antigens, the serum titers from a mouse immunized with one of the five recombinant V-antigen proteins were measured by ELISAs. Cluster analysis of the OD values from the ELISA was performed, from which phylogenetic trees and a heat map were constructed (**Fig 4B**). LssV and AcrV showed higher cross-

(OD 450 nm)

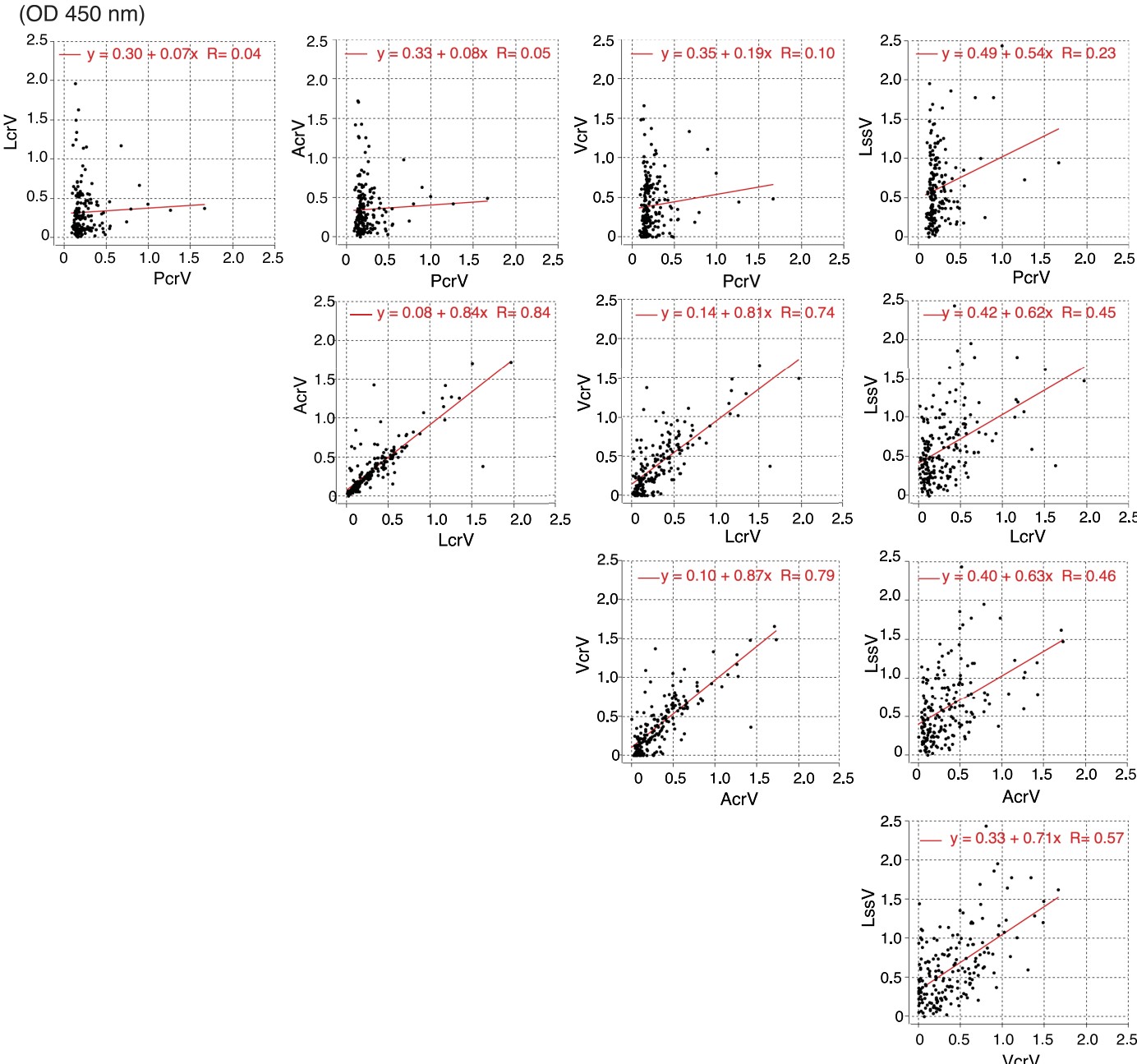

**Fig 3. Serum titer correlations between two V-antigens.** The diluted serum (1,280×) was applied to ELISAs, and OD values at 450 nm were measured. The titer correlation of the two V-antigens was mapped in an X–Y plot. OD: Optical density.

antigenicity with each other, unlike VcrV that was less cross-reactive with the other V-antigens (Fig 4B). Whereas the sera from the AcrV-immunized mice reacted with PcrV, the sera from PcrV-immunized mice did not react with AcrV.

Next, to determine whether correlations among the V-antigens had some association with the identity of the primary protein sequences, we investigated the identity of these sequences in the five V-antigens using the BLOSUM substitution score matrix in ClustalW (Fig 6). In this map, the overall similarities of two out of five of the V-antigens for the whole molecules

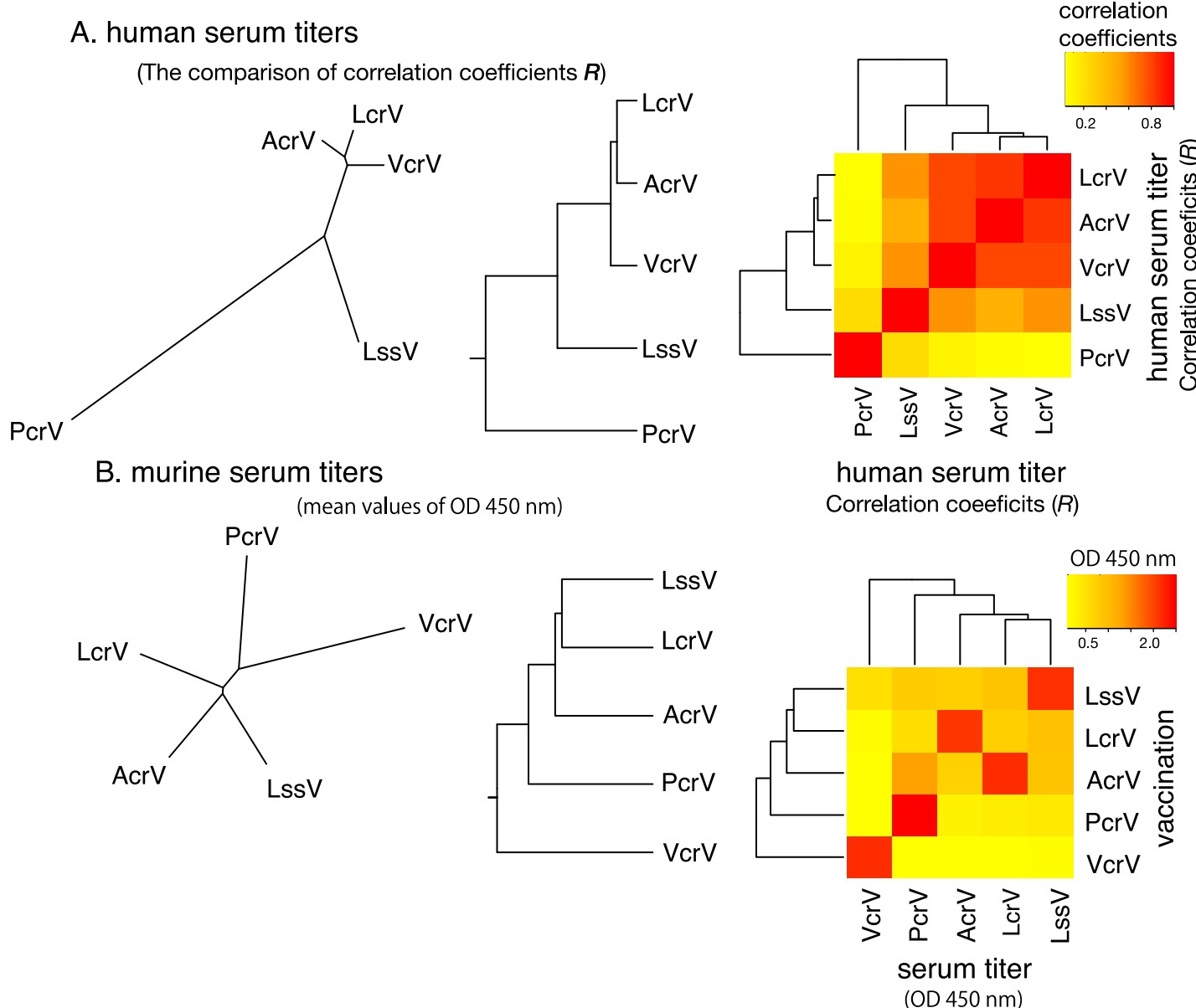

**Fig 4. Phylogenetic trees and heat maps showing correlations among the five anti-V-antigen titers. A.** Human serum titers. The correlation coefficients of the serum titer correlations shown in **Fig 3** were matrixed. Phylogenetic trees (an unrooted tree, neighbor-joining method, and a rooted tree, unweighted pair group method with arithmetic mean) and a heat map demonstrating correlation coefficient values of the serum titer plots (Fig 3) as a color-index were constructed. **B.** Anti-V-antigen serum titers from mice immunized with one of the five V-antigens. The mean values from triplicate measurements were applied to construct a phylogenetic tree (unrooted tree, neighbor-joining method, and a rooted tree, unweighted pair group method with the arithmetic mean) and a heat map demonstrating O.D 450 nm values of the serum titers to the V-antigens as a color-index. OD: Optical density.

ranged between 21% and 49%. The similarity of the amino-terminal domain (14%–51%) and central domains (14%–45%) was low in comparison with the carboxyl-terminal domain that showed 48%–84% similarity. VcrV was unique with a long additional sequence (>160 amino acids, aa) at the amino-terminal domain and an additional sequence of 80 aa in the central domain of the sequence alignment.

We also constructed phylogenetic trees of the V-antigens based on the primary amino acid sequence similarity scores of whole molecules, amino-terminal domains, central domains, and

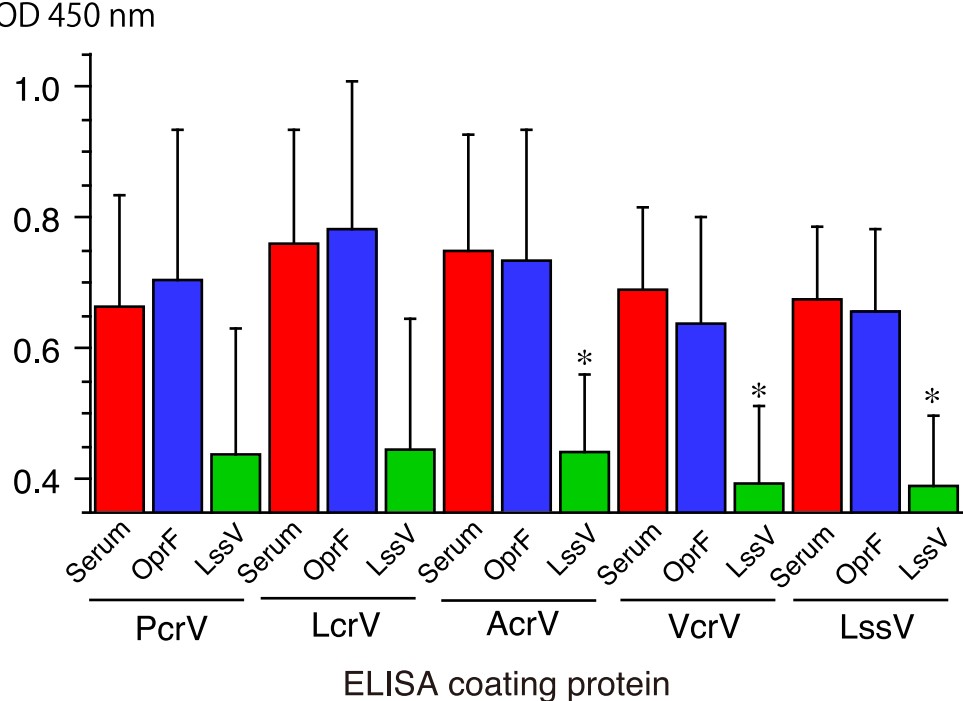

**Fig 5. Inhibition ELISA.** Anti-V-antigen titers of human sera (n = 2), which showed relatively high titers against all five V-antigens, preincubated with either recombinant LssV or OprF were measured in comparison with the original titer levels of the sera. $^*p < 0.05$ with the Mann-Whitney U-test between mean values of original serum titers and those of titers after pre-inhibition with recombinant proteins. OD: Optical density.

carboxyl-terminal domains (**Fig 7**). In the cluster analysis of whole molecules, the similarity score was the highest between LcrV and LssV (48.6%) (**Fig 7A**). The amino-terminal domains showed similarity scores of <30%, except for the 51.9% similarity between LcrV and LssV (**Fig 7B**). For the center domains, the highest similarity score was obtained between AcrV and LssV (44.5%) (**Fig 7C**). In the cluster analysis of the carboxyl-terminal domains with the highest similarity scores (48%–84%) for their amino acid sequences, AcrV, LssV, and PcrV were closer than VcrV and LcrV (**Fig 7D**).

These phylogenetic analyses using the primary amino acid sequences did not match well with the phylogenetic trees constructed from the correlations among the serum IgG titers of the five V-antigens (**Fig 4A**). In particular, VcrV, which has an extremely long central domain containing regions that are missing in the other V-antigens, occupied a separate position in the phylogenetic trees constructed from the primary amino acid sequences. Therefore, these findings suggest the titer cross-antigenicity in human sera may not be correlated with the similarity in primary amino acid sequences.

As we have reported previously, a blocking monoclonal antibody called Mab166 against PcrV recognizes the conformational structure, but not the primary amino acid sequence [15]. As shown in **Fig 8**, the predicted three-dimensional structures of the V-antigens had similar dumbbell-like structures with two globular domains on either end of a grip formed by two coiled-coil motifs [37]. The grip that connected the two globular domains contained an antiparallel coiled-coil structure comprising central and carboxyl-terminal coiled-coil regions. The carboxyl-terminal was folded into a single long α-helix. Regarding the blocking antibody recognizing the three-dimensional conformational epitopes and not the primary amino acid sequences, not only the serum titer levels, but also the specific components that bind to the

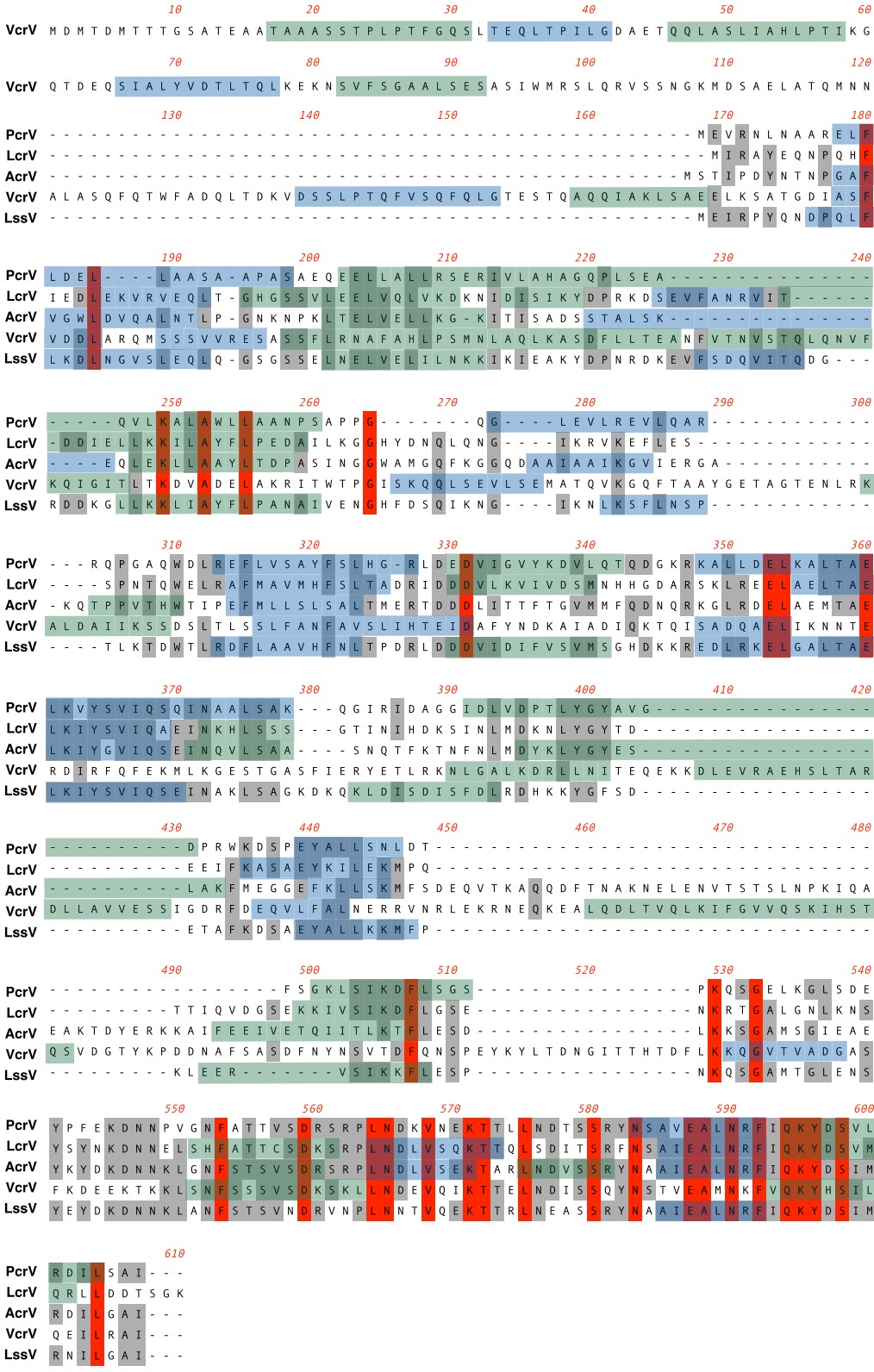

**Fig 6. ClustalW sequence alignment of the five primary V-antigen sequences.** The overall sequence similarities of two of the five V-antigens across the whole molecules were between 21% and 49%. The similarity of the amino-terminal domain (14%–51%) and central domains (14%–45%) was low in comparison with the carboxyl-terminal domain that

had a 48%–84% similarity range. VcrV was unique with a long additional amino-terminal domain sequence (>160 aa) and an additional sequence (80 aa) in the central domain of the sequence alignment.

specific blocking regions in V-antigen molecules may be important to prevent the pathogenesis associated with TTSS virulence.

## Discussion

*Yersinia* LcrV has been recognized as a V-antigen with immunoprotective characteristics in *Yersinia* infections since the 1950s [4, 5, 7, 8]. However, after almost 50 years, LcrV, *P. aeruginosa* PcrV, and *Aeromonas* AcrV were anatomically visualized as distinct structures on the tip of the needle of the injectisome of the type III secretion apparatus [38–41]. Because specific antibodies binding to a particular portion of this structure inhibit the translocation of type III secretory toxins in *Yersinia* and *P. aeruginosa* [11, 42], gaining a better understanding of the interactions between V-antigens and the host humoral immunity against the virulence of bacterial type III secretion is important to develop potential non-antibiotic treatments for infections in various hosts.

Cross-antigenicity among *Yersinia* spp. was reported nearly 40 years ago. In 1980, cross-immunity to *Y. pestis* was noted in mice that had been orally infected with *Y. enterocolitica* serotype O3 [43]. In 1983, it was also reported that partial protection in mice against *Y. pestis*

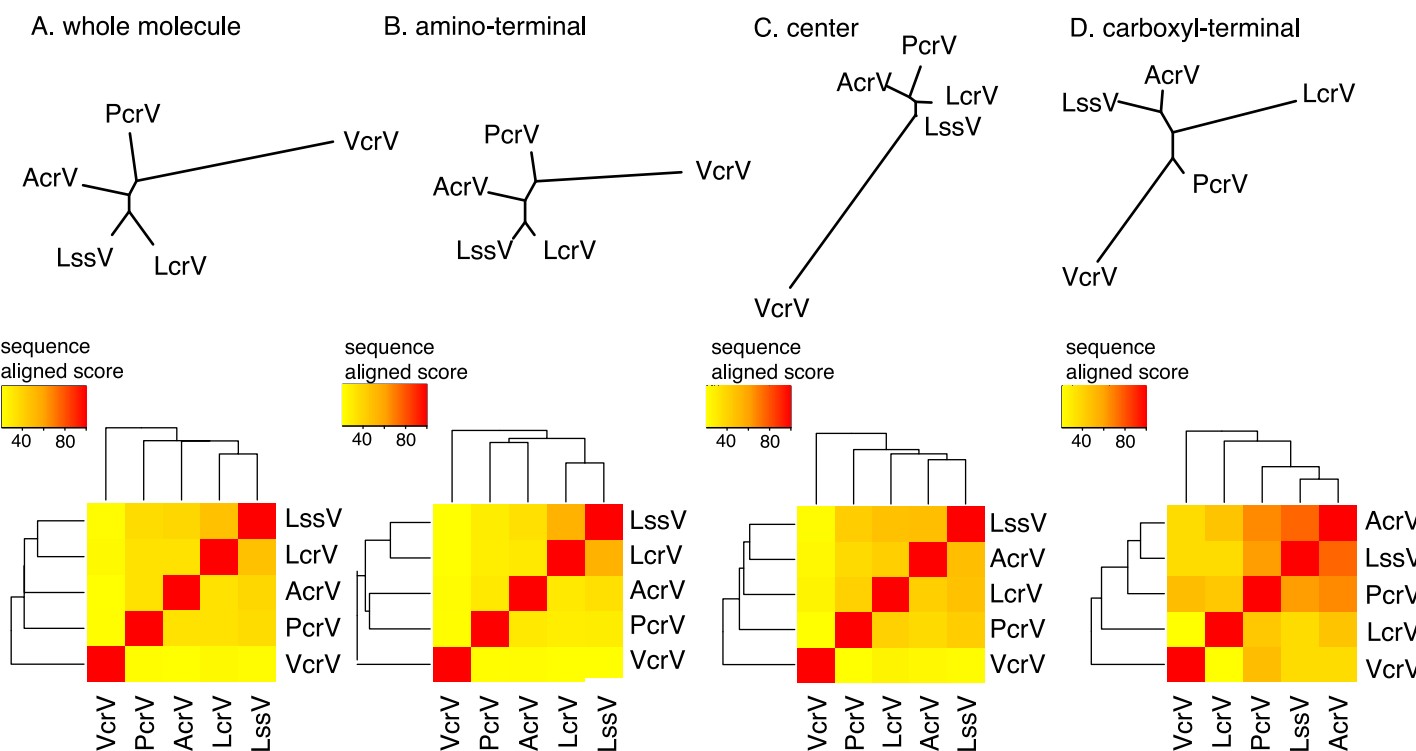

**Fig 7. Phylogenetic trees and heat maps based on the primary amino acid sequences (whole molecule, amino-terminal, center, and carboxyl-terminal) of the V-antigens.** Phylogenetic analyses of the primary amino acid sequences of whole molecules, amino-terminal, center, and carboxyl-terminal domains were performed. The sequence alignment scores obtained from ClustalW are shown. Phylogenetic trees (an unrooted tree, neighbor-joining method) and a heat map demonstrating sequence alignment scores between two V-antigens as a color-index were constructed. A. Complete primary sequence. B. Amino-terminal domain, C. Center domain, D. Carboxyl-terminal domain. The amino acid positions in the amino-terminal, center, and carboxyl-terminal domains are as follows: LcrV: #1–#164, #165–#278, and #279–#326 respectively; PcrV: #1–#142, #143–#256, and #257–#294, respectively; AcrV: #1–#162, #163–#316, and #317–#361, respectively; VcrV: #1–#361, #362–#558, and #559–#607, respectively; LssV: #1–#170, #171–#280, and #281–#325, respectively.

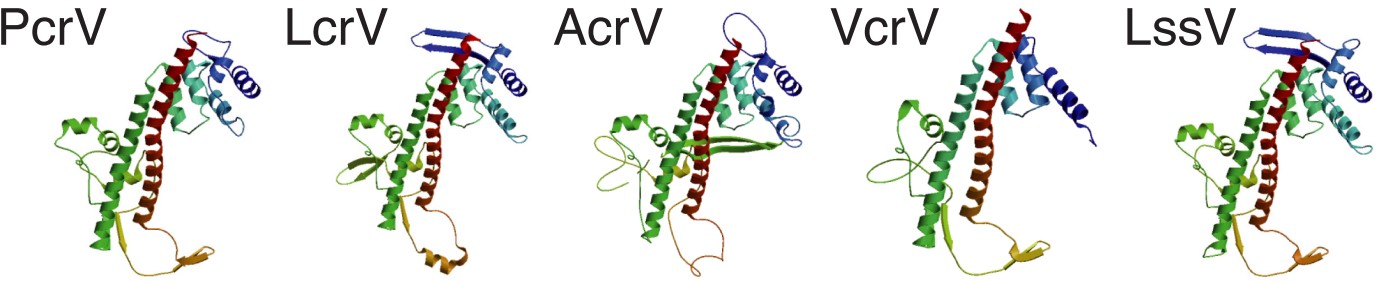

**Fig 8. Predicted three-dimensional V-antigen structures.** The predicted three-dimensional structures were generated by the Cn3D macromolecular structure viewer at the National Center for Biotechnology Information (https://www.ncbi.nlm.nih.gov/Structure/CN3D/cn3d.shtml).

infection by the *Y. enterocolitica* V-antigen was linked to the partial cross-reactivity of V-antigens [44]. Later, DNA sequencing of the most common serotypes of human pathogenic *Y. enterocolitica* and *Y. pseudotuberculosis* revealed two evolutionary distinct types of V-antigen in *Yersinia* spp. [45]. One type is represented by *Y. enterocolitica* serotype O8 strains WA, WA-314, and NCTC 10938 (LcrV-YenO8), whereas the other type is represented by *Y. pestis*, *Y. pseudotuberculosis*, and *Y. enterocolitica* serotypes O3, O9, and O5 (LcrV-Yps) [45]. By raising monospecific antisera against both types of V-antigen (*Y. enterocolitica* serotypes O3 and O8), anti-V-antigen serum was protective only when the immunizing V-antigen was the same type as the V-antigen produced by the infecting strain. The difference in protective immunity between the two types was caused by the presence of a hypervariable region between amino acids 225 and 232 [45]. The protectivity of the V-antigen was later confirmed and refined using the recombinant V-antigen of *Y. pseudotuberculosis* and monospecific anti-V-antigen serum [46]. In this previous study, antiserum against the *Y. pseudotuberculosis* V-antigen provided mice with passive immunity to challenge infection with *Y. pestis* or *Y. pseudotuberculosis*, but not *Y. enterocolitica* O8 (strain WA). These previous studies on cross-antigenicity to *Yersinia* V-antigens in mice imply that the structure of the critical domain, but not the overall primary amino acid sequence similarity *per* se, is important for protective immunity.

Other than *Yersinia* LcrV, only two studies of *P. aeruginosa* PcrV have reported antibody titers against *P. aeruginosa* PcrV in human serum [32, 47]. No other studies investigating antibody titers against other V-antigens have been reported to date. *V. parahaemolyticus*, a Gram-negative marine bacterium, of which the V-antigen titers were examined in the present study, causes food-borne gastroenteritis [48, 49]. Among *Vibrio* spp., *V. harveyi* (a Gram-negative bioluminescent marine bacterium) is ubiquitous in the marine environment. It is considered as one of the important bacterial species that form the normal flora of healthy shrimp, and its genome carries a V-antigen homolog gene [33, 50]. This bacterium is sometimes recognized as causing high mortality of shrimps in the worldwide shrimp fishing industry [51, 52]. As a halophilic *Vibrio* species, *V. alginolyticus*, which causes wound infections, was first recognized as pathogenic to humans in 1973 [53]. Recent studies have proposed that *V. alginolyticus* possesses the same TTSS gene organization as *V. parahaemolyticus* and *V. harveyi* [54, 55]. *Aeromonas* spp., such as *A. salmonicida* and *A. hydrophila*, are not pathogenic to humans, but cause infections in salmon and trout, and carry the AcrV V-antigen homolog [33]. However, *A. hydrophila* sometimes causes gastroenteritis in humans who acquire such infections by ingesting food or water containing sufficient numbers of this organism [56–58]. *Ph. luminescens* (a gammaproteobacterium in the *Enterobacteriaceae* family) is also not pathogenic to humans, but is a lethal pathogen of insects and possesses a pathogenicity island encoding the TTSS including the LssV V-antigen homolog [59–62]. It lives in the gut of an

entomopathogenic nematode in the *Heterorhabditidae* family [33]. However, human infections with *Photorhabdus* spp. have recently been reported in the USA and Australia, suggesting that these bacteria are emerging human pathogens [63]. In comparison with the infections caused by *P. aeruginosa*, infections caused by *Y. pestis* or *V. parahaemolyticus* are less common. Therefore, it is interesting that titers against V-antigens from non-human pathogens were also detected in this study. Anti-AcrV showed a high correlation with anti-LcrV, and some correlation was detected between anti-LssV and anti-VcrV. The titers against the antigens from non-human pathogens might result from cross-antigenicity among the V-antigen homologs. Our cluster analysis with the heat map of anti-V-antigen titers from the 186 adult volunteers displayed a higher correlation between LcrV, AcrV, and VcrV than the correlation between PcrV and LssV and the other V-antigens (**Fig 4A**).

Immunizing mice with one of the five recombinant V-antigens resulted in cross-antigenicity of the sera against the V-antigens (**Figs 4B** and **5**). This result shows that VcrV is unique, while AcrV, LssV, and LcrV share some degree of cross-antigenicity with each other, although the results differed slightly from the cross-antigenicity observed with human sera. In our previous study on anti-PcrV titers in human sera, administration of extracted IgG derived from high titer anti-PcrV sera protected against challenge with lethal pneumonia in a murine model [18]. In this study, among the 198 volunteer-derived sera [32], the top 10 high titer sera against PcrV (greater than or equivalent to 0.5 at OD 450 nm) were used. Although we have human monoclonal IgG specific against PcrV as a standard for anti-PcrV measurement, no such human standard IgG against the other four V-antigens are available to date. It is difficult at this time to clearly evaluate where the level of protection is sufficient, but as many as 5% of the volunteers showed protective levels in our previous study [18], and the cross-antigenicity among the five V-antigens probably has a certain level of clinical significance in human immunity. Therefore, despite no similar immunological experiments being performed with other V-antigens, our study has shown that the cross-reactive antigenicity we observed among the various V-antigens with serum-specific anti-V antigen titers may afford some degree of immunological protection against various Gram-negative bacteria. Further investigation of the conformational blocking epitopes in the needle cap structure of type III secretory apparatus and the immunological aspects of the structural antigenicity of a critical portion of the V-antigens should provide a better understanding of how to effectively block the TTSS-associated virulence associated with various Gram-negative pathogens.

## Supporting information

**S1 Fig.**
(JPEG)

## Acknowledgments

We thank Sandra Cheesman, Ph.D., and Mitchell Arico from Edanz Group (www.edanzediting.com/ac) for editing a draft of this manuscript.

## Author Contributions

**Conceptualization:** Teiji Sawa.

**Data curation:** Mao Kinoshita, Teiji Sawa.

**Formal analysis:** Mao Kinoshita, Teiji Sawa.

**Funding acquisition:** Teiji Sawa.

**Investigation:** Mao Kinoshita, Teiji Sawa.

**Methodology:** Mao Kinoshita, Masaru Shimizu, Koichi Akiyama, Hideya Kato.

**Project administration:** Teiji Sawa.

**Resources:** Masaru Shimizu, Kiyoshi Moriyama.

**Supervision:** Teiji Sawa.

**Writing – original draft:** Mao Kinoshita, Teiji Sawa.

**Writing – review & editing:** Kiyoshi Moriyama, Teiji Sawa.

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
