## [Decision Letter · Decision Letter 0]

12 Nov 2019

PONE-D-19-20754

Epidemiological survey of serum titers from adults against various Gram-negative bacterial V-antigens

PLOS ONE

Dear Dr. Sawa,

Thank you for submitting your manuscript to PLOS ONE. After careful consideration, we feel that it has merit but does not fully meet PLOS ONE’s publication criteria as it currently stands. Therefore, we invite you to submit a revised version of the manuscript that addresses the points raised during the review process.

Please review closely the comments of the three reviewers. I agree with the reviewers that the experimental methods and figures need to be edited for clarity and more controls to be implemented.

I hope you find the comments useful to guide you to significantly improve the manuscript.

We would appreciate receiving your revised manuscript by Dec 27 2019 11:59PM. To enhance the reproducibility of your results, we recommend that if applicable you deposit your laboratory protocols in protocols.io, where a protocol can be assigned its own identifier (DOI) such that it can be cited independently in the future. For instructions see: http://journals.plos.org/plosone/s/submission-guidelines#loc-laboratory-protocols

We look forward to receiving your revised manuscript.

Kind regards,

Katerina Kourentzi, PhD

Academic Editor

PLOS ONE

Journal Requirements:

3. Please provide additional details regarding participant consent. In the ethics statement in the Methods and online submission information, please ensure that you have specified (1) whether consent was suitably informed and (2) what type you obtained (for instance, written or verbal). If your study included minors under age 18, state whether you obtained consent from parents or guardians. If the need for consent was waived by the ethics committee, please include this information.

4. To comply with PLOS ONE submissions requirements, please provide methods of sacrifice in the Methods section of your manuscript.

5. We noticed you have some minor occurrence(s) of overlapping text with the following previous publication(s), which needs to be addressed:

https://doi.org/10.1111/1348-0421.12353

https://doi.org/10.1111/1348-0421.12147

https://iai.asm.org/content/iai/65/2/446.full.pdf

In your revision ensure you cite all your sources (including your own works), and quote or rephrase any duplicated text outside the Methods section. Further consideration is dependent on these concerns being addressed.

6. Thank you for stating the following in the Competing Interests section:

I have read the journal's policy and the authors of this manuscript have the following competing interests: T. Sawa has patents associated with PcrV immunization (World Patent No. WO0033872; European Patent No. EP1049488; U.S. Patent No. 6309651; U.S. Patent No. 6827935, and Japan Patent No. 2017-020501). Until 2011, T. Sawa received a patent fee from the Regents of the University of California related to the development of a therapeutic monoclonal antibody at KaloBios Pharmaceutical. There are no current financial relationships existing with any organization associated with this study.

Reviewers' comments:

Reviewer's Responses to Questions

**Comments to the Author**

1. Is the manuscript technically sound, and do the data support the conclusions?

Reviewer #1: Partly

Reviewer #2: Partly

Reviewer #3: Partly

2. Has the statistical analysis been performed appropriately and rigorously? 

Reviewer #1: No

Reviewer #2: I Don't Know

Reviewer #3: Yes

3. Have the authors made all data underlying the findings in their manuscript fully available?

Reviewer #1: Yes

Reviewer #2: Yes

Reviewer #3: Yes

4. Is the manuscript presented in an intelligible fashion and written in standard English?

Reviewer #1: Yes

Reviewer #2: Yes

Reviewer #3: Yes

5. Review Comments to the Author

Reviewer #1: In this manuscript, Kinoshita et al tried to establish serum anti-V antibody titer as method for the epidemiological investigations on various pathogenic Gram-negative bacteria. Authors developed an enzyme-linked immuno-sorbent assay to measure titers against each V antigen in the sera collected from 186 adult volunteers. However, the manuscript is identified with a rather confusing experimental approaches for epidemiological investigation of Gram-negative bacterial pathogens. The application of purified V antigens (PcrV from Pseudomonas aeruginosa, LcrV from Yersinia, LssV from Photorhabdus luminescens, AcrV from Aeromonas salmonicida and VcrV from Vibrio parahaemolyticus) and their cross reactivity in both human and mouse serum samples demonstrating the inability of this system to determine accurately the bacterium elicited antigenicity in human samples. The manuscript is also suffering with the complete lack of standard or control samples in entire experiments and that raised serious question against reproducibility of work.

Authors must include the closely related species and or strains of all five bacteria used in this study to validate it further.

Reviewer #2: Major comments:

1) Figure 4A doesn't match to the written part that describe it and actually it is identical to figure 7A. Please correct the figure.

2) There is no statistical method section in the Materials and Methods part so it is impossible to determine statistical significance, power analysis etc. Please include such a section with the relevant statistical analyses.

3) Regarding the heat maps, please provide a valid statistical/numerical information that defines the various colors. I'm sure that this data is provided by the relevant analysis tools. Please include a short description also in the figure legend.

Minor comments:

1)The sentence in line 273 is not clear.

2)Line 30; a dot is missing after the 0.7

3)Line 117; please correct the 0.0M to a valid value.

4)Line 119; please correct the 200mL to 200μL

5)Line 121; please correct the 100mL/well to 100μL/well

6)line 125; please correct the 100mL/well to 100μL/well

7)line 224; please correct "..carboxyl domains.." to carboxyl-terminal domains

Reviewer #3: This paper by Kinoshita et. al. looks into the antibody response to various bacterial V-antigens in a cohort of healthy adults. They find that many people have various antibody responses to the V-antigens and that many of these antibodies may cross-react. The data is sound and the authors are careful not to overstate their findings. I do have a few comments however;

1. It is not made clear what percent of people have reactive antibodies to each protein. This could be a percent of patients over a specified cutoff. You already have both ‘red’ and ‘orange’ cutoffs so these could be used.

2. Both the correlations in Figure 3 and the mice vaccination studies strongly suggest the presence of some cross-reactive antibodies in the human sera. This would be fairly easy to prove in the human sera by absorption experiments. Take a human sera with responses to multiple proteins- adsorb against the highest reacting protein and determine if this also removes response to the other V-antigens.

3. Figure 5 would be easier to interpret if it was a graph or heat map of the response levels instead of a photo of the plate. I can’t tell by eye if one well is more blue than another.

4. Lines 244-248 the authors state that the amino-acid sequences phylogeny in Fig 7 do not match the serum trees in Fig 4A. To my eye they do match pretty well especially 7A and 7B. In both LssV is closest to LcrV and VcrV is furthest away. Can the authors clarify their statement?

6. PLOS authors have the option to publish the peer review history of their article (what does this mean?). If published, this will include your full peer review and any attached files.

Reviewer #1: No

Reviewer #2: No

Reviewer #3: No

---

## [Author Response · Author response to Decision Letter 0]

22 Dec 2019

Journal Requirements:

[Response]

I followed the above guidance.

[Response]

We provide an original uncropped and unadjusted SDS-PAGE image of Fig.1, and wrote it in our cover letter for this resubmission.

3. Please provide additional details regarding participant consent. In the ethics statement in the Methods and online submission information, please ensure that you have specified (1) whether consent was suitably informed and (2) what type you obtained (for instance, written or verbal). If your study included minors under age 18, state whether you obtained consent from parents or guardians. If the need for consent was waived by the ethics committee, please include this information.

[Response]

We added the following:

Page 8, line 104 (page 8, line 139 of tracked_change_version): “As a non-interventional and non-invasive retrospective observational study, the need for consent was waived by the ethics committee.”

4. To comply with PLOS ONE submissions requirements, please provide methods of sacrifice in the Methods section of your manuscript.

[Response]

We added the following:

Page 11, line 152 (page 11, line 214 of tracked_change_version):” …., the immunized mice were euthanized with a high dose intraperitoneal injection of sodium pentobarbital,”

5. We noticed you have some minor occurrence(s) of overlapping text with the following previous publication(s), which needs to be addressed:

https://doi.org/10.1111/1348-0421.12353

https://doi.org/10.1111/1348-0421.12147

https://iai.asm.org/content/iai/65/2/446.full.pdf

In your revision ensure you cite all your sources (including your own works), and quote or rephrase any duplicated text outside the Methods section. Further consideration is dependent on these concerns being addressed.

[Response]

We provided the correct reference numbers in all the sentences which we cited.

6. Thank you for stating the following in the Competing Interests section:

I have read the journal's policy and the authors of this manuscript have the following competing interests: T. Sawa has patents associated with PcrV immunization (World Patent No. WO0033872; European Patent No. EP1049488; U.S. Patent No. 6309651; U.S. Patent No. 6827935, and Japan Patent No. 2017-020501). Until 2011, T. Sawa received a patent fee from the Regents of the University of California related to the development of a therapeutic monoclonal antibody at KaloBios Pharmaceutical. There are no current financial relationships existing with any organization associated with this study.

[Response]

We added the sentence as follows, in the section Conflict of Interest:

This does not alter our adherence to PLOS ONE policies on sharing data and materials.

Responses to Reviewer comments:

Reviewer #1: In this manuscript, Kinoshita et al tried to establish serum anti-V antibody titer as method for the epidemiological investigations on various pathogenic Gram-negative bacteria. Authors developed an enzyme-linked immuno-sorbent assay to measure titers against each V antigen in the sera collected from 186 adult volunteers. However, the manuscript is identified with a rather confusing experimental approaches for epidemiological investigation of Gram-negative bacterial pathogens. The application of purified V antigens (PcrV from Pseudomonas aeruginosa, LcrV from Yersinia, LssV from Photorhabdus luminescens, AcrV from Aeromonas salmonicida and VcrV from Vibrio parahaemolyticus) and their cross reactivity in both human and mouse serum samples demonstrating the inability of this system to determine accurately the bacterium elicited antigenicity in human samples. The manuscript is also suffering with the complete lack of standard or control samples in entire experiments and that raised serious question against reproducibility of work.

Authors must include the closely related species and or strains of all five bacteria used in this study to validate it further.

[Response]

We understand some of the concerns raised by the reviewer. There is no standard human anti-V-antigen IgG except for human monoclonal anti-PcrV IgG. Therefore, after optimization of the ELISA system for anti-PcrV titers using the anti-PcrV monoclonal standard [ref 32], the optical density measured under a consistent condition with the same secondary antibody was used to evaluate the titers. We do not agree with the reviewer’s suggestion to use closely related species or strains of all five bacteria. We used five recombinant V-antigens from five different species. Among the same species, variation in the primary sequences of V-antigens is quite low. Thus, there is no reason to include closely related species or strains. However, in an additional experiment, we included P. aeruginosa OprF as an irrelevant non-V-antigen protein that has no similarity with the tested V-antigens in terms of primary sequences. As a result, in the inhibition ELISA using OprF, preincubation with OprF did not affect the optical density measured for anti-V-antigen titers, whereas preincubation with LssV significantly decreased anti-V-antigen titers for ArcV and LssV as evidence of cross-antigenicity (new Fig. 5 Inhibition ELISA). 

As the reviewer pointed out, there is no standard sample for titer measurement of anti-LcrV, anti-LssV, anti-VcrV, and anti-AcrV. Except for human monoclonal anti-PcrV IgG mAb 6F5 as a standard to measure the anti-PcrV titer [32], there is no human standard anti-V-antigen IgG. Therefore, after optimization of the ELISA system for anti-PcrV titers using the mAb 6F5 standard [32], the optical density measured under a consistent condition with the same secondary antibody was used to evaluate the titers. Accordingly, we have added the following text.

Page 14, line 202 (page 13, line 282 of tracked_change_version): Next, the inhibition ELISA was performed to show cross-antigenicity among V-antigens. As an irrelevant non-crossreactive protein, we prepared P. aeruginosa recombinant OprF using the same E.coli-derived recombinant protein construction system. The sequence alignment scores obtained from ClustalW between OprF and V-antigens were 9.8–11.4, whereas those among V-antigens were from 21.3 (between VcrV and AcrV) to 48.3 (between LcrV and LssV). Three human sera diluted 1000× were preincubated with either recombinant LssV or recombinant OprF overnight. Then, preprocessed serum titers against each V-antigen were measured in comparison with the titer levels of the original sera (Fig. 5). As a result, preincubation with OrpF did not affect the specific titer levels. However, preincubation with LssV decreased the titer levels compared with the titer levels of the original sera (*p<0.05 for AcrV, VcrV, and LssV). 

Page 9, line 130 (page 9, line 178 of tracked_change_version): Except for human monoclonal anti-PcrV IgG mAb 6F5 as a standard to measure the anti-PcrV titer [32], there is no human anti-V-antigen IgG. Therefore, after optimization of the ELISA system for anti-PcrV titers using the mAb 6F5 standard [32], the OD measured under a consistent condition with the same secondary antibody was used to evaluate the titers.

Reviewer #2: Major comments:

1) Figure 4A doesn't match to the written part that describe it and actually it is identical to figure 7A. Please correct the figure.

[Response]

We have corrected the error pointed out by the reviewer. In the revised manuscript, Figure 4A is the correct figure. 

2) There is no statistical method section in the Materials and Methods part so it is impossible to determine statistical significance, power analysis etc. Please include such a section with the relevant statistical analyses.

[Response]

As suggested, we have added a new section “Statistical analysis” in the Materials and methods.

3) Regarding the heat maps, please provide a valid statistical/numerical information that defines the various colors. I'm sure that this data is provided by the relevant analysis tools. Please include a short description also in the figure legend.

[Response]

We have added numerical information about the color heat maps in figures. We have added the methodology to construct the heat maps in the newly added section “Phylogenetic and cluster analyses“ and the explanation of the color-index in each figure legend.

Minor comments:

1)The sentence in line 273 is not clear.

[Response]

As pointed out by the reviewer, the description of this sentence was unclear. We have rewritten the sentence as follows.

Page 20, line 316 (page 20, line 525 of tracked_change_version): Regarding the blocking antibody recognizing the three-dimensional conformational epitopes and not the primary amino acid sequences, not only the serum titer levels, but also the specific components that bind to the specific blocking regions in V-antigen molecules may be important to prevent the pathogenesis associated with TTSS virulence. 

2)Line 30; a dot is missing after the 0.7

[Response]

The error has been corrected.

3)Line 117; please correct the 0.0M to a valid value.

It has been changed to “0.05 M”.

4)Line 119; please correct the 200mL to 200μL

[Response]

It has been changed to “200 μL”.

5)Line 121; please correct the 100mL/well to 100μL/well

[Response]

It has been changed to “100 μL”.

6)line 125; please correct the 100mL/well to 100μL/well

[Response]

It has been changed to “100 μL”.

7)line 224; please correct "..carboxyl domains.." to carboxyl-terminal domains

[Response]

The error has been corrected.

Reviewer #3: This paper by Kinoshita et. al. looks into the antibody response to various bacterial V-antigens in a cohort of healthy adults. They find that many people have various antibody responses to the V-antigens and that many of these antibodies may cross-react. The data is sound and the authors are careful not to overstate their findings. I do have a few comments however;

1. It is not made clear what percent of people have reactive antibodies to each protein. This could be a percent of patients over a specified cutoff. You already have both ‘red’ and ‘orange’ cutoffs so these could be used.

[Response]

On the basis of our previous results published in ref 18, we have added our speculation about the sufficient levels at which we can anticipate the protective effects because it is difficult to determine the cut-off levels.

Page 24, line 398 (page 24, line 640 of tracked_change_version): In this study, among the 198 volunteer-derived sera [32], the top 10 high titer sera against PcrV (greater than or equivalent to 0.5 at OD 450 nm) were used. Although we have human monoclonal IgG specific against PcrV as a standard for anti-PcrV measurement, no such human standard IgG against the other four V-antigens are available to date. It is difficult at this time to clearly evaluate where the level of protection is sufficient, but as many as 5% of the volunteers showed protective levels in our previous study [18], and the cross-antigenicity among the five V-antigens probably has a certain level of clinical significance in human immunity.

2. Both the correlations in Figure 3 and the mice vaccination studies strongly suggest the presence of some cross-reactive antibodies in the human sera. This would be fairly easy to prove in the human sera by absorption experiments. Take a human sera with responses to multiple proteins- adsorb against the highest reacting protein and determine if this also removes response to the other V-antigens.

[Response]

As the reviewer suggested, we have performed the additional experiment. We conducted an inhibition ELISA (sometimes called a competitive ELISA) to distinguish the levels of specific binding from the levels due to non-specific protein bindings. For high titer sera, we preincubated with either recombinant OprF (a Pseudomonas aeuriginosa surface antigen protein that is irrelevant to the V-antigen) or recombinant LssV and then measured the specific anti-V-antigen titers. The results are shown in new Fig. 5.

Page 14, line 202 (page 13, line282 of tracked_change_version): Next, the inhibition ELISA was performed to show cross-antigenicity among V-antigens. As an irrelevant non-crossreactive protein, we prepared P. aeruginosa recombinant OprF using the same E.coli-derived recombinant protein construction system. The sequence alignment scores obtained from ClustalW between OprF and V-antigens were 9.8–11.4, whereas those among V-antigens were from 21.3 (between VcrV and AcrV) to 48.3 (between LcrV and LssV). Three human sera diluted 1000× were preincubated with either recombinant LssV or recombinant OprF overnight. Then, preprocessed serum titers against each V-antigen were measured in comparison with the titer levels of the original sera (Fig. 5). As a result, preincubation with OrpF did not affect the specific titer levels. However, preincubation with LssV decreased the titer levels compared with the titer levels of the original sera (*p<0.05 for AcrV, VcrV, and LssV). 

3. Figure 5 would be easier to interpret if it was a graph or heat map of the response levels instead of a photo of the plate. I can’t tell by eye if one well is more blue than another.

[Response]

In accordance with the reviewer comment, we have omitted the previous Fig. 5.

4. Lines 244-248 the authors state that the amino-acid sequences phylogeny in Fig 7 do not match the serum trees in Fig 4A. To my eye they do match pretty well especially 7A and 7B. In both LssV is closest to LcrV and VcrV is furthest away. Can the authors clarify their statement?

[Response]

We apologize for our error. In Fig 4A, we mistakenly used Fig. 7A. We have corrected the error.

Comments

In this manuscript, Kinoshita et al tried to establish serum anti-V antibody titer as method for the epidemiological investigations on various pathogenic Gram-negative bacteria. Authors developed an enzyme-linked immuno-sorbent assay to measure titers against each V antigen in the sera collected from 186 adult volunteers. However, the manuscript is identified with a rather confusing experimental approaches for epidemiological investigation of Gram-negative bacterial pathogens. The application of purified V antigens (PcrV from Pseudomonas aeruginosa, LcrV from Yersinia, LssV from Photorhabdus luminescens, AcrV from Aeromonas salmonicida and VcrV from Vibrio parahaemolyticus) and their cross reactivity in both human and mouse serum samples demonstrating the inability of this system to determine accurately the bacterium elicited antigenicity in human samples. The manuscript is also suffering with the complete lack of standard or control sample in entire experiments and that raised serious question against reproducibility of work. 

[Response]

As the reviewer pointed out, there is no standard sample for titer measurement of anti-LcrV, anti-LssV, anti-VcrV, and anti-AcrV. Except for human monoclonal anti-PcrV IgG mAb 6F5 as a standard to measure the anti-PcrV titer [32], there is no human standard anti-V-antigen IgG. Therefore, after optimization of the ELISA system for anti-PcrV titers using the mAb 6F5 standard [32], the optical density measured under a consistent condition with the same secondary antibody was used to evaluate the titers. We added the description about it in Page 9, line 130, as follows:

“Except for human monoclonal anti-PcrV IgG mAb 6F5 as a standard to measure the anti-PcrV titer [32], there is no human anti-V-antigen IgG. Therefore, after optimization of the ELISA system for anti-PcrV titers using the mAb 6F5 standard [32], the OD measured under a consistent condition with the same secondary antibody was used to evaluate the titers.”

Accordingly, we have added the following text. In addition, to confirm the cross-antigenicity, we added an additional experiment (inhibition ELISA with irrelevant protein OprF), and demonstrated the result in the new Fig 5.

---

## [Decision Letter · Decision Letter 1]

29 Jan 2020

PONE-D-19-20754R1

Epidemiological survey of serum titers from adults against various Gram-negative bacterial V-antigens

PLOS ONE

Dear Dr. Sawa,

Thank you for submitting your  revised manuscript to PLOS ONE. After careful consideration, we feel that it has merit but does not fully meet PLOS ONE’s publication criteria as it currently stands. Before accepting it for publication I would like to ask you to carefully review and clarify/fix the remaining small issues (scientific and grammatical) as noted by the reviewers.

Therefore, we invite you to submit a revised version of the manuscript that addresses the points raised during the review process.

We would appreciate receiving your revised manuscript by Mar 14 2020 11:59PM. To enhance the reproducibility of your results, we recommend that if applicable you deposit your laboratory protocols in protocols.io, where a protocol can be assigned its own identifier (DOI) such that it can be cited independently in the future. For instructions see: http://journals.plos.org/plosone/s/submission-guidelines#loc-laboratory-protocols

We look forward to receiving your revised manuscript.

Kind regards,

Katerina Kourentzi, PhD

Academic Editor

PLOS ONE

Additional Editor Comments (if provided):

Thank you for your revised manuscript!

Before accepting it for publication I would like to ask you to carefully review and clarify/fix the remaining small issues (scientific and grammatical) as noted by the reviewers.

Reviewers' comments:

Reviewer's Responses to Questions

**Comments to the Author**

1. If the authors have adequately addressed your comments raised in a previous round of review and you feel that this manuscript is now acceptable for publication, you may indicate that here to bypass the “Comments to the Author” section, enter your conflict of interest statement in the “Confidential to Editor” section, and submit your "Accept" recommendation.

Reviewer #1: All comments have been addressed

Reviewer #2: All comments have been addressed

Reviewer #3: All comments have been addressed

2. Is the manuscript technically sound, and do the data support the conclusions?

Reviewer #1: Partly

Reviewer #2: Yes

Reviewer #3: Yes

3. Has the statistical analysis been performed appropriately and rigorously? 

Reviewer #1: Yes

Reviewer #2: Yes

Reviewer #3: Yes

4. Have the authors made all data underlying the findings in their manuscript fully available?

Reviewer #1: Yes

Reviewer #2: Yes

Reviewer #3: Yes

5. Is the manuscript presented in an intelligible fashion and written in standard English?

Reviewer #1: Yes

Reviewer #2: Yes

Reviewer #3: Yes

6. Review Comments to the Author

Reviewer #1: Authors have modified the manuscript as per the comments. However, revised manuscript is still having few grammatical mistakes, which need to be addressed before final submission.

Reviewer #2: I acknowledge the authors for addressing my remarks. Few points that still need attention.

In lines 163, 165 and 166 of the revised manuscript there are empty parentheses that either need to be filled or should be deleted.

In figure 4B the order of the V antigen of the 5 species on the two axes is not the same. Is there a reason for that? If the answer is no please correct the order.

Reviewer #3: As stated above all my comments have been addressed sufficiently. The new Figure 5 improves the manuscipt especially.

7. PLOS authors have the option to publish the peer review history of their article (what does this mean?). If published, this will include your full peer review and any attached files.

Reviewer #1: No

Reviewer #2: No

Reviewer #3: No

---

## [Author Response · Author response to Decision Letter 1]

2 Feb 2020

Review Comments to the Author

Reviewer #1: Authors have modified the manuscript as per the comments. However, revised manuscript is still having few grammatical mistakes, which need to be addressed before final submission.

[Response to the comment]

We fixed the following grammatical errors.

Page 4, line 50: the virulence associated -> the virulence-associated

Page 6, line 74: correlated and -> correlated, and

Page 6, line 87: multicloning site -> multiple cloning site

Page 8, line 108: March 2013 participated -> March 2013, participated

Page 9, line 128: a typo error: inhibiton -> inhibition

Page 11, line 153: by ELISAs as described -> by ELISAs, as described

Page 11, line 161: method and rooted trees -> method, and rooted trees

Page 12, line 176: regression analysis -> a regression analysis

Page 12, line 180: as considered statistically significant. -> considered statistically significant.

Page 13, line 204: coefficient 0.84 -> coefficient of 0.84

Page 14, line 226: unique profile -> a unique profile

Page 15, line 235: method -> method,

Page 15, line 240: method -> method,

Page 17, line 278: additional sequence -> an additional sequence

Page 22, line 357: a challenge infection -> challenge infection

Page 22, line 369: shrimp and -> shrimp, and

Page 23, line 380: Luminescens -> luminescens

Page 25, line 414: the type III secretory apparatus -> type III secretory apparatus

Page 25, line 416: better understanding -> a better understanding

Page 26, line 431: Science and -> Science, and

Page 26, line 433: PhD, -> Ph.D.,

Reviewer #2: I acknowledge the authors for addressing my remarks. Few points that still need attention.

In lines 163, 165 and 166 of the revised manuscript there are empty parentheses that either need to be filled or should be deleted.

[Response to the comment]

Page 11, line 162- 165 fixed (Actually, there are not empty parentheses, but the R function(). However, we eliminated the parentheses to avoid the confusion)

In figure 4B the order of the V antigen of the 5 species on the two axes is not the same. Is there a reason for that? If the answer is no please correct the order.

[Response to the comment]

We fixed the error. Now, we aligned them in a correct order.

Reviewer #3: As stated above all my comments have been addressed sufficiently. The new Figure 5 improves the manuscript especially.

[Response to the comment]

We thank you very much for your good advice!

---

## [Editor Report · Decision Letter 2]

25 Feb 2020

Epidemiological survey of serum titers from adults against various Gram-negative bacterial V-antigens

PONE-D-19-20754R2

Dear Dr. Sawa,

We are pleased to inform you that your manuscript has been judged scientifically suitable for publication and will be formally accepted for publication once it complies with all outstanding technical requirements.

With kind regards,

Katerina Kourentzi, PhD

Academic Editor

PLOS ONE
---

## [Editor Report · Acceptance letter]

27 Feb 2020

PONE-D-19-20754R2 

Epidemiological survey of serum titers from adults against various Gram-negative bacterial V-antigens 

Dear Dr. Sawa:

I am pleased to inform you that your manuscript has been deemed suitable for publication in PLOS ONE. Congratulations! Your manuscript is now with our production department. 

With kind regards,

on behalf of

Dr. Katerina Kourentzi 

Academic Editor

PLOS ONE